# Identifying unrecognised risks to life from debris flows

Mark Bloomberg[1*], Tim Davies[1], Elena Moltchanova[1], Tom Robinson[1], and David Palmer[2]

[1]University of Canterbury, Private Bag 4800 Christchurch 8140, New Zealand

[2]Scion, PO Box 29237, Riccarton, Christchurch 8440, New Zealand

* Corresponding author.

**Abstract.** Many debris-flow catchments pose an underappreciated hazard, especially where there are dwellings on debris-flow fans and other depositional areas. There is a need to make communities and those involved in community governance aware of situations where there

may be a credible risk to life from debris flows. This needs to be simple and cheap to do, since funding is often not available to study unrecognised natural hazards. Here, we use published models to 1) estimate the threshold annual recurrence interval (ARI) for debris flows in a catchment, below which there is an unacceptable annual risk to life for the occupiers of any dwellings, and 2) identify the "window of non-recognition" where debris

flows are sufficiently infrequent within a catchment that it is not recognised as susceptible, yet frequent enough that the risk to life exceeds the acceptability threshold.
Using four New Zealand studies, we estimate a 95% credible interval range for the ARIs of life-threatening debris flows of between 100 and 500 years. We show that, given these credible intervals and precautionary but realistic assumptions about debris flow behaviour

and the vulnerability of dwellings and their occupants, catchments with no history of debris flow activity can pose an unrecognised and unacceptable annual risk to life ($P$=0.256 that the annual risk to life threshold of 1 in 1000 is exceeded).

# 1 Introduction

Debris-flows are intense sediment-flood events that can occur in steep, erodible catchments when heavy rainfall causes slope failures to deliver large quantities of fine sediment to stream channels (Jakob et al., 2005). This input then causes sediments to be mobilised in the channel as discrete surge waves containing boulders and often trees that move rapidly down-channel to fan areas, where they can be destructive and potentially fatal (Iverson, 2014). New Zealand

is prone to such events because of its active tectonic, volcanic and hydrological setting and many steep, erodible catchments (Welsh and Davies, 2011; Farrell and Davies, 2019). Debris flows are often unrecognised and underappreciated by the New Zealand public (Welsh and Davies, 2011). This is partly due to confusing terminology, with debris flows referred to as "floods", "flash floods", or "slips" (McSaveney et al., 2005). However, the behaviour and

impacts of debris flows are very different from conventional floods or landslips on a hillside. For the same amount of rain, a debris flow can have a much higher instantaneous discharge rate, contain much more and often much larger rock debris, and move faster than a flood in the same location (Jakob and Jordan, 2001). Therefore, in a given catchment, debris flows are usually far more hazardous and harder to manage than floods (Dowling and Santi, 2014;

McSaveney et al., 2005). At the same time, their flow behaviour means they can travel very large distances, impacting environments far from their sources (Frank et al., 2015). In contrast, potentially catastrophic slips and other landslides generally occur on steep slopes, and their impacts occur within a limited zone downslope from the landslide.

*Problem statement and objectives*
There is a large and growing literature on debris-flow hazard assessments (Jakob, 2021), but implementing these assessments requires funding. Thus, the debris flow literature has an inherent bias towards relatively complex studies involving a range of site assessment and modelling techniques. There is a lack of studies that describe how to overcome the problem

described by Jakob (2021): "Most districts, states, provinces, or even nations have limited funds for geohazard mitigation. This necessitates the allocation of existing funds to those sites with the highest risk potential. Funds for studies and mitigation often get allocated because of particularly damaging events that result in focused public, media, and political attention. Those sites, however, may not necessarily be the ones with highest risk."

Although there are catchments that generate debris flows with average recurrence intervals (ARIs) of a few years or less (see Davies et al., 2024, Table 1 for a summary), many have

ARIs ranging from decades to millennia (Jakob, 2005). Consequently, many debris flow-susceptible catchments have no record of debris flow activity, resulting in an underappreciated hazard.

The primary requirement for a debris flow to occur is a large volume of sediment, especially fine sediment, available for mobilisation by a triggering event. This requires a steep and erodible catchment so that hillslope processes can deliver sediment to the stream channel (Welsh and Davies, 2010). Thus, catchment gradient is an obvious factor likely to be associated with debris flow occurrence, and numerous morphometric indices for debris flow

susceptibility have been proposed based on catchment topography (de Haas et al., 2024). The most-used indicator variable is the Melton ratio ($R$), which measures a catchment's average steepness (Melton 1965). $R$ is calculated from:

$$R = H/(A^{0.5}) \tag{1}$$

where $A$ is the map area of the catchment surface ($m^2$), and $H$ is the elevation difference

between the catchment's highest point and fan apex (m).

Various studies have derived a range of threshold values for $R$, above which a catchment is deemed susceptible to debris flows. A typical threshold for debris-flow susceptibility is $R$ >0.5 or >0.6 (Holm et al. 2016, Page et al. 2012, Welsh and Davies 2010, Wilford et al. 2004). However, in practice, there is no well-defined $R$ threshold, with debris flows occurring

in catchments with $R$ values down to as low as 0.15 (e.g. Davies et al., 2024; Church and Jakob, 2020; McSaveney et al., 2005).

Morphometric indices such as the Melton $R$ have proved useful for regional-scale assessment of debris flow susceptibility using geospatial analysis in both Europe and North America (e.g. Bertrand et al., 2017; Cavalli et al., 2017; Holm et al., 2016; Ilinca, 2021). In New Zealand,

regional-scale mapping of catchment $R$ (Welsh and Davies, 2010; Bloomberg and Palmer, 2022) suggests that significant areas of built environments in New Zealand may be subject to debris flow hazards, even where no previous events have been recorded. However, these regional reconnaissance-level studies require follow-up by agencies responsible for natural hazard management, i.e., detailed site investigation of potential debris flow hazards and risks

at the site level.

Of particular concern are locations where debris flows pose a risk to life for occupants of dwellings on debris flow fans. The lack of a quantified ARI makes accurately calculating risk difficult. In this case, "…unquantified (or ignored) risks can lead to incomplete or irrational risk management" decisions (Strouth and McDougall, 2022).

Here, we describe a simple method to rapidly and easily estimate the annual risk that debris flows pose to dwellings located on debris flow fans and, thus, the annual risk to life for the occupiers of those dwellings. We utilise these methods to show that even though debris flows may have ARIs of centuries, their ability to cause great damage means the risk they pose to life can exceed acceptable levels. Nonetheless, the long ARIs for these events create an

illusory sense of security so that their risk to life is not recognised.

## 2  Methods

### 2.1 Setting acceptable limits to risk to life from potential debris flow hazards

Globally, the individual risk to life from natural hazard impacts is considered unacceptable at

levels greater than about $10^{-3}$ to $10^{-4}$ per year (Taig et al., 2012). Where multiple deaths can occur, graphs showing the expected frequency and cumulative number of fatalities (F-N curves) can indicate societal risk and its tolerability (e.g. Fig. 1 in Porter and Morgenstern (2012)). Such graphs are widely used as indicators of acceptable risk limits for various hazards but vary in the thresholds for acceptable risk (Mona, 2014; Sim et al., 2022). Here, we

follow Porter and Morgenstern (2012) to establish a maximum acceptable individual risk to life as $10^{-3}$ per year, which scales linearly with the maximum acceptable risk to multiple lives ($10^{-3}/N$ per year, where N= number of fatalities).

*Calculation of risk to life for a debris flow catchment*

If the Melton $R$ or other evidence suggests that a catchment may be susceptible to debris flows, and there are existing or proposed dwellings on the debris flow fan, then there is a need to demonstrate to communities and those involved in community governance that there may be a risk to life from debris flows. This demonstration needs to be credible yet simple and inexpensive since funding is often not available to study unrecognised natural hazards.

To achieve this, we use a modified form of a commonly used calculation of the annual risk to life from exposure to a single landside (see Walker et al., 2007; Jakob et al., 2012; Porter and Morgenstern, 2022; de Vilder et al., 2022):

$$R_{DF} = P_H * P_{S:H} * P_{T:S} * V * E \qquad (2)$$

where: $R_{DF}$ is the individual risk to life from a debris flow event; $P_H$ = annual probability of the debris flow occurring; $P_{S:H}$ = spatial probability of impact on a dwelling if a debris flow

occurs; $P_{T:S}$ = temporal probability that an individual occupant will be present when the debris flow impacts the dwelling; $V$ = vulnerability, or probability of loss of life if the occupied dwelling is impacted; and $E$ = number of occupants at risk, which is equal to 1 for the determination of individual risk. $P_H$ can also be specified in terms of its inverse, the average recurrence interval (ARI, years) for a debris flow event.

We retain the notation but redefine some of the variables in Eq. 2 to reflect our understanding of the components of risk to life from debris flows. We redefine the "Risk" term as the maximum acceptable annual risk to life ($R_{DF}(max)$) and $P_H$ as $P_H(max)$, the value for the annual probability of a debris flow that will result in $R_{DF}(max)$, so that:

$$R_{DF}(max) = P_H(max) * P_{S:H} * P_{T:S} * V * E \qquad (3)$$

$$P_H(max) = R_{DF}(max) / [P_{S:H} * P_{T:S} * V * E] \qquad (4)$$

Eq. 4 allows $P_H(max)$ to be calculated, given an accepted value for $R_{DF}(max)$ and known or assumed values for $P_{S:H}$, $P_{T:S}$, $V$ and $E$. If there is evidence that the annual probability of a debris flow occurring is greater than the calculated $P_H(max)$, then any occupants of dwellings on the debris flow fan will be subject to an unacceptable risk to life.

Eq. 4 also allows us to explore the effects of uncertainty about the values of its other parameters. These parameters and their uncertainties are discussed in the following sections.

*Probability of impact on a dwelling if a debris flow occurs ($P_{S:H}$)*

If a debris flow occurs, it will likely discharge onto a debris flow fan, typically a depositional area where a steepland catchment disgorges onto a lower-slope landform. Initially, the debris flow is likely to follow existing active stream channels. However, changes in the active-channel position, termed avulsions, can pose a severe threat to dwellings on fans. This is because mitigation measures (e.g., check dams, bunds) are usually applied to active channels and cannot prevent damage from flows that establish a new channel pathway (de Haas et al., 2018). Thus, a dwelling on the fan can still be impacted, even if it is far from existing stream channels.

Furthermore, the path(s) followed by the avulsing debris flows are difficult to predict (de Haas et al., 2018). A very conservative assumption is, therefore, $P_{S:H}$ =1. In other cases where debris fans may be small or truncated by wave action or river flows, dwellings are often sited on the fan apex and $P_{S:H}$ =1 is near-certain.

*Temporal probability that an individual will be present when the landslide occurs $P_{T:S}$*

In New Zealand, the average proportion of time an individual spends within a residential dwelling is 0.69 (Khajehzadeh and Vale, 2017). However, this average value may not apply during high-intensity rainfall events when debris flows are most likely. At such times, dwelling occupants may self-evacuate or be evacuated by the authorities. Alternatively, during the event, the proportion of time spent in the dwelling may be close to 1, as the occupants shelter in place. We use a value of $P_{T:S}$ =0.69 for this study, recognising that actual values are likely to be binomial (one or zero) during high-intensity rainfall events.

*Probability of an individual death if dwelling impact occurs (V)*

This parameter is critical but has considerable uncertainty. Firstly, it depends on debris flow intensity in terms of volume, depth, composition and velocity. While somewhat governed by catchment area and topography, debris flow volumes may vary by at least two orders of magnitude between the median and 99[th] percentile for catchments of the same area (de Haas and Densmore, 2019; Marchi et al., 2019). Other factors (rainfall intensity and the volume of landslide material available for mobilisation as debris flows) are difficult to estimate or predict but likely to be important drivers of debris flow intensity (e.g. Chang et al. 2011). Also important to the value of the *V* parameter is the vulnerability of the impacted building since casualties in landslides are often related to the destruction of occupied buildings and are thus a function of structural vulnerability (Jakob et al., 2012; Pollock and Wartman, 2020). Massey et al. (2018) review building vulnerability studies and state that building performance during impact from a natural hazard depends on the type of structure or "building typology". To describe the susceptibility of a building to damage from landslide hazards, most authors use the building typology. For example, Kang and Kim (2016) analysed data from 11 debris flow events in different parts of South Korea in July/August 2011. All events resulted in damage to buildings from debris flow impacts. For these events, vulnerability functions were related to the debris flow depth, flow velocity, and impact pressure. Separate vulnerability functions were estimated for reinforced concrete frame buildings and non-reinforced concrete frame buildings, with reinforced concrete frame buildings having much lower vulnerability. Finally, *V* may depend on chance, timing or human behaviour. For example, out of caution, occupants may move to a less vulnerable part of the dwelling during a high-intensity rainfall event (Pollock and Wartman, 2020). Conversely, if the debris flow occurs in the middle of

the night, a person sleeping in a bedroom on the upslope side of a dwelling may have no warning or chance to avoid the full force of impact.

We assume that the risk of death for an individual in an impacted dwelling is $V = 0.1$. This is consistent with Bell and Glade (2004), who published values for the risk of death to an individual within a building (0.02 to 0.25) for "low-magnitude" to "high-magnitude" debris flow events, respectively—although they did not specifically define the terms "low-magnitude" or "high-magnitude". Note that if we chose a value of $V=1.0$ (it is certain the occupant of an impacted dwelling would die), then the threshold for $P_H(\text{max})$ would be an order of magnitude lower, assuming we use the same threshold $R_{DF}(\text{max})$ (0.001 in this study).

*The number of occupants at risk (E)*

The $P_H(\text{max})$ value is based on the $R_{DF}(\text{max})$ value for the number of occupants in a single dwelling. Note that the maximum acceptable annual probability of debris flows $P_H(\text{max})$ becomes progressively smaller with increasing $E$. In other words, the risk to life will increase with an increasing number of dwellings (and therefore $E$), and the acceptable-risk threshold for the annual probability of debris flows should be reduced. At the same time, some factors may reduce the risk to life with increasing $E$. If the larger numbers of people $E$ are dispersed over multiple dwellings on a fan, and if the debris flow path and deposition area are restricted to part of the fan, some dwellings may not be impacted. Thus, the decrease in $P_H(\text{max})$ might not scale linearly with increasing $E$ since $P_{S:H}$ is less if averaged over all the dwellings. We wish to avoid this complexity as the parameter values in Eq. (2) will vary amongst different dwellings located on a debris flow fan. For simplicity, we assume we are estimating the risk to life from a debris flow event for an individual in a dwelling subject to the highest risk. However, we also assume that other individuals in the same dwelling will have a similar risk. Therefore, we used the NZ occupancy rate for usually resident households ($N_O=2.67$, Statistics NZ (2023)) to calculate the number of occupants at risk *(E)* for a single dwelling on a debris flow fan. This approach was also used by Bell and Glade (2004), who estimated an individual risk to a person in a building and then multiplied this by the total number of occupants in the building to estimate an "Object risk to people in buildings", defined as the risk to life taking all people at a building into account.

**2.2 Analysis of potential risk to life**

*The estimated annual probability of debris flow occurrence ($P_H$)*

Calculating $P_H$(max) provides a standard for comparison with estimated $P_H$ (annual probability of a debris flow occurring). There is an unacceptable risk to life for debris flow catchments where estimated $P_H$ > calculated $P_H$(max). However, estimated $P_H$ can have wide confidence limits or be completely uncertain since ARIs may be centuries or even millennia in magnitude. This means that no debris flows may have occurred in living or even historical

memory for most catchments, so data to estimate ARIs are sparse or lacking.

This lack of certainty is a serious problem since 1) $P_H$ is an important driver of annual risk to life from debris flows, 2) the lack of observations means estimates of $P_H$ may have confidence limits that are so wide as to make the estimates uninformative, and 3) in rapidly-developing countries like New Zealand, the expansion of land use into hitherto-unutilised

areas means that debris-flow hazard may be unrecognised. Of course, with very long ARIs (very low $P_H$), the risk to life may be acceptably low. However, there may be a "window of non-recognition" where ARIs are long enough that the debris flow hazard is not yet recognised but short enough that the risk to life is still unacceptably high. The second application of our model is identifying any such window.

Our model assumes a single annual probability threshold $P_H$(max) for a debris flow that results in an unacceptable risk to life for occupiers of a dwelling in a debris flow catchment. A more complex formulation would account for the reality that debris flows of different magnitudes/intensities may come from the same catchment, with larger, more intense events having lower frequencies. For example, Strouth and McDougall (2022) estimate separate

model parameter values for each frequency-magnitude scenario, then integrate these to estimate an overall risk to life. However, this requires sufficient data to estimate frequency-magnitude relationships (Jakob et al., 2020). As pointed out earlier in this paper, our method is designed for situations where there may be no data on debris flow occurrences since either 1) none will have occurred within recorded history or 2) funding was not available to carry

out the required study.

*Using Bayesian analysis to estimate $P_H$*

We used Bayesian inference to estimate $P_H$ for debris flow events. In a previous study, we used the upper bound of $P_H$ values from studies of known debris flow catchments (see Table

1, Davies et al., 2024) to estimate the risk to life from debris flow hazards. However, this

approach has the disadvantage that $P_H$ values will be based on the most active debris flow catchments, leading to the criticism that any estimates of risk to life are overly pessimistic ("risk estimate conservatism"), which is to be avoided in evaluating risk (Strouth et al., 2024).

Bayesian analysis uses expert opinion to estimate a "prior" distribution of the variable of interest (in this case, $P_H$) combined with any available observed evidence to produce a "posterior" distribution. This has the advantage that it accounts for the full range of catchment $P_H$ values, not just the values for the most active catchments.

There are few formal expert estimates of ARI or $P_H$ for debris flows in New Zealand
catchments. Table 1 summarises ARIs for four well-studied debris flow catchments–Awatarariki Stream, Mātata (McSaveney et al., 2005), Karaka Stream, Thames township (McSaveney and Beetham, 2006), Nyhane Drive, Ligar Bay (Page et al., 2012) and Brewery Creek, Queenstown (Beca Ltd, 2020). ARIs are for debris flows that observation or modelling suggested were potentially life-threatening.

**Table 1. ARI and estimated sizes of four well-studied debris flow catchments in New Zealand. Size classes are according to Jakob (2005).**

| Name and date | Size Class | Volume (m³) | Estimated ARI (years) |
|---|---|---|---|
| Awatarariki Stream, Mātata | 5 | 200,000 | 200-500 |
| Karaka Stream, Thames | 5 | $10^5$-$10^6$ | Midrange $10^2$-$10^6$ |
| Ligar Bay[1] | - | Not reported | 200-500 |
| Brewery Creek, Queenstown [2] | 3 | 1580-5560 | 50-200 |
|  | 4 | 10,410-16,685 | 200-2500 |
|  | 4-5 | 98,330-139,300 | 2500-10,000 |

[1] Page et al. (2012) noted that the estimated 200-year ARI for a debris flow catchment in Ligar Bay may be reduced, possibly by up to half, based on climate-change projections.
[2] ARIs were based on simulations for three debris flow magnitudes (small-medium-large). The smallest magnitude (ARI 50-200 years) still resulted in an unacceptable risk to life near the top of the fan apex.

Based on Table 1, we used two conjugate beta-binomial models with the beta prior $P_H \sim$ Beta ($a$, $b$), where the parameters $a$ and $b$ were chosen to correspond to the prior 95% credible
intervals for $P_H$ of (1/500,1/200) and (1/500,1/100) respectively. We then assume that there have been no observed life-threatening debris flows in a catchment for 100 years. This "observation" allows us to estimate the posterior probability and 95% credible intervals for $P_H$ for that catchment. We then compared the estimated $P_H$ with the $P_H$(max) values,

assuming one dwelling per catchment. Where estimated $P_H > P_H(max)$, the risk to life was

classified as unacceptable.

We also estimated the probability of a "window of non-recognition" where ARIs are long enough that the debris flow hazard is not recognised but short enough that the risk to life is still unacceptably high. For three defined periods (50, 100 and 150 years), we estimated the posterior predictive distributions for the probabilities of outcomes where zero debris flows

occurred during the period since, for these outcomes, the debris flow susceptibility of the catchment would likely be unrecognised (assuming no expert investigation of the catchment). This assumes that if at least one debris flow had occurred in a catchment during the specified period, it would have been recorded, and the catchment's susceptibility would have been clearly recognised.

Note that while we have used 50, 100 and 150-year periods, these methods can be used with any period. The criterion for the choice of period is how far back it is likely that a debris flow occurrence would be remembered and recorded. In regions where human settlement is very recent, an appropriate period might be considerably shorter than 150 years.

All statistical analysis was done in programme R (R Core Team, 2021).

### 3    Results

### 3.1 Estimation of $P_H(max)$ and comparison with estimated $P_H$

Table 2 shows the calculated values for $P_H(max)$ assuming the upper limit for acceptable risk to an individual life ($R_{DF}(max)$) is 0.001, probability of impact $P_{S:H} = 1.0$ and probability of

individual death if a debris flow impacts an occupied dwelling ($V = 0.1$). It also shows the inverse of $P_H(max)$, the minimum ARI threshold below which the risk to life is unacceptable.

**Table 2. Parameters used to calculate $P_H(max)$, the maximum acceptable annual probability of a debris flow occurring.**

| Parameter | Symbol | Value |
|---|---|---|
| Maximum annual acceptable individual risk to life | $R_{DF}(max)$ | 0.001 |
| Probability of impact on a dwelling if a debris flow occurs | $P_{S:H}$ | 1.0 |
| Probability of individual death if a dwelling impact occurs | $V$ | 0.1 |
| Average no of occupants per dwelling (NZ residential) | $N_O$ | 2.67 |
| Average proportion of time that the dwelling is occupied | $P_{T:S}$ | 0.69 |
| Dwellings/catchment | | 1 |
| Individuals/catchment ($N_O*$ no of dwellings) | $E$ | 2.67 |
| Maximum acceptable annual debris flow probability | $P_H(max)$ | 0.00543 |
| Minimum acceptable debris flow ARI (years) | | 184 |

The upper acceptable threshold for the annual probability of a debris flow ($P_H(max)$) can be used to explore the risk to life from debris flows by comparing it with estimated $P_H$ from Bayesian inference. Unacceptable risk to life occurs where the estimated annual probability of a debris flow $P_H$ exceeds the $P_H(max)$ threshold in Table 2 (0.00543).

Table 3 summarises the parameters for the Bayesian estimates. Prior estimates are for two possible ranges for ARI, 200–500 years and 100–500 years. Posterior estimates are based on the assumed "observation" that no life-threatening debris flows have occurred within the last 100 years.

**Table 3. Parameters for Bayesian estimates of debris flow $P_H$.**

| Parameter | Parameter value | |
|---|---|---|
| 95% intervals for $P_H$ | 1/200, 1/500 | 1/100, 1/500 |
| Prior coefficients ($a$, $b$) | 19.41, 6198.57 | 6.72, 1499.8 |
| Posterior coefficients ($a$, $b$) | 19.41, 6298.57 | 6.72, 1599.8 |
| Posterior probability $P_H > P_H(max)$ | 0.060 | 0.2560 |

Figure 1 shows how $P_H(max)$ for a single dwelling can be compared with 1) the prior distribution of $P_H$ and 2) a posterior distribution of $P_H$ that assumes zero observations in a catchment for a 100-year period.

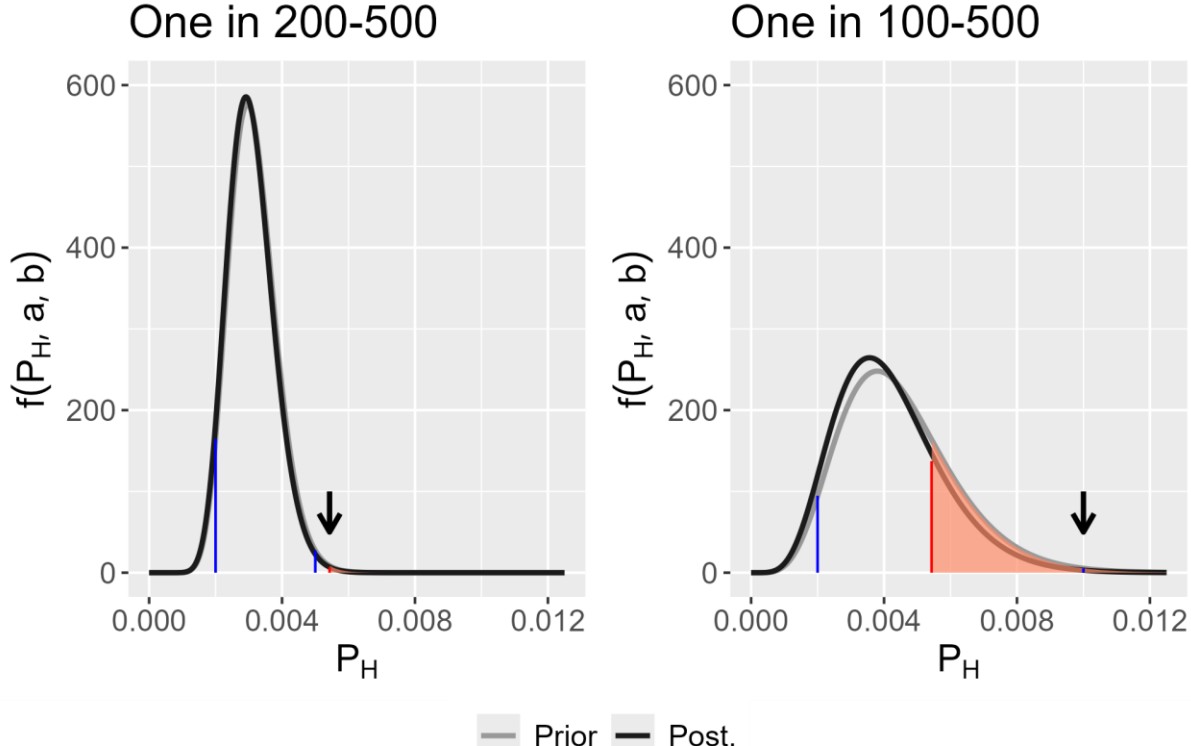

**Figure 1 Prior (grey) and posterior (black) probability distribution for $P_H$, assuming the 95% limits for $P_H$ are (left-hand graph) (1/200, 1/500) or (right-hand graph) (1/100, 1/500). Posterior probabilities are calculated with zero occurrences of debris flows over 100 years. The orange area under the curve corresponds to the posterior probability that $P_H > P_H(max)$ for a single dwelling. $P_H(max) = 0.00543$ is shown by a vertical red line. The blue vertical lines indicate the values for the prior probabilities. The arrows are used to indicate probability lines that are too small to see: $P_H(max)$ in the left hand graph, the prior probability one in 100 (0.01) in the right hand graph.**

Using the posterior distributions in Fig. 1, the posterior probability that $P_H > P_H(max)$ is the area under the black curve to the right of the red vertical line ($P_H(max) = 0.00543$ for a single dwelling) (Table 2). For the 95% credible intervals for $P_H$ of (1/500,1/200), the posterior probability = 0.0060; therefore, it is highly unlikely that $P_H > P_H(max)$. For the 95%

credible intervals for $P_H$ of (1/500,1/100), the posterior probability that $P_H > P_H(max)$ = 0.2560. In this case, there is a reasonably high probability that the maximum acceptable risk to life $P_H(max)$ would be exceeded.

**3.2 Estimating the "window of non-recognition"**

The probability distributions for the probability of zero events in 50, 100 and 150 years were used to identify a "window of non-recognition" where ARIs are long enough that the debris flow hazard is not recognised but short enough that risk to life is still unacceptably high. These distributions are based on the credible intervals (1/500, 1/200) or (1/500, 1/100) for the $P_H$ of a life-threatening debris flow event in a catchment (Table 4, Figure 2).

**Table 4. Probability of zero life-threatening debris flow events within a nominated period, using two priors assuming the 95% limits for $P_H$ are (1/200, 1/500) or (1/100, 1/500). CI= the associated 95% credible interval for the mean posterior predicted probabilities.**

| Assumptions | Probabilities | |
|---|---|---|
| 95% credible intervals for $P_H$ | 1/200, 1/500 | 1/100, 1/500 |
| Zero events in 50 years | 0.8559 | 0.8026 |
| (95% CI) | (0.7919,0.9101) | (0.6554,0.9162) |
| Zero events in 100 years | 0.7335 | 0.6486 |
| (95% CI) | (0.6271,0.8283) | (0.4296,0.8394) |
| Zero events in 150 years | 0.6293 | 0.5277 |
| (95% CI) | (0.4966,0.7538) | (0.2816,0.7690) |

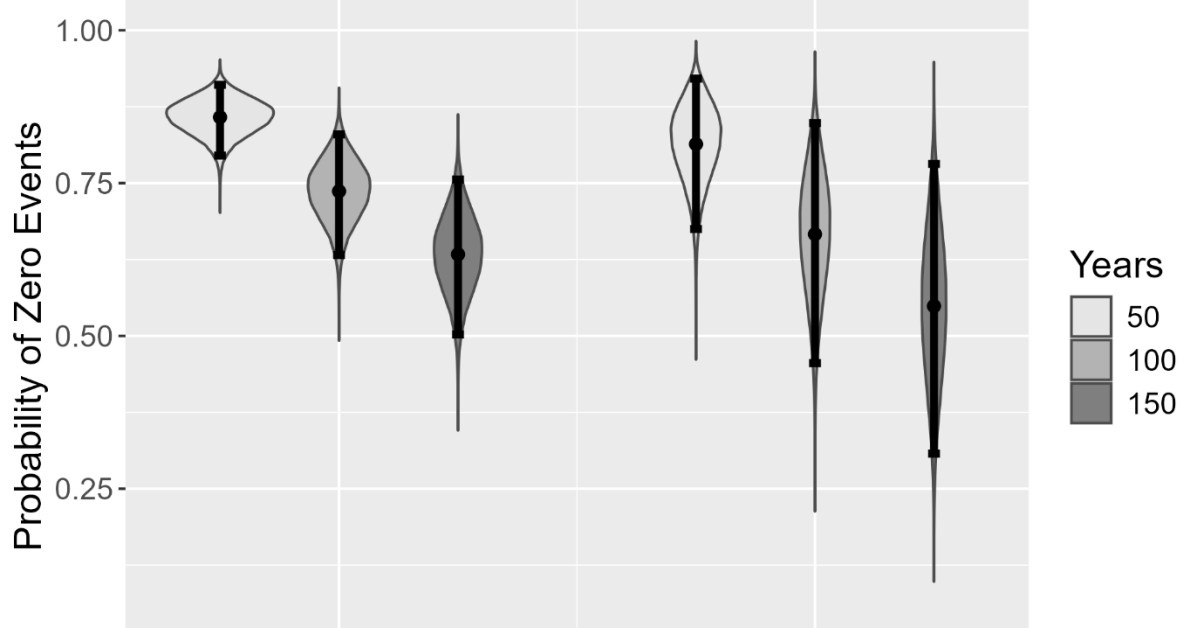

**Figure 2 The mean posterior predicted probabilities and the underlying densities (violin plots) for zero events in 50, 100 and 150 years, assuming $P_H$ estimated with credible intervals of (1/500, 1/200) (left-hand graph) or (1/500, 1/100) (right-hand graph). Error bars are the associated 95% credible interval for the mean posterior predicted probabilities.**

If we use 95% credible intervals for $P_H$ (1/500, 1/100), the mean probability that no life-threatening debris flow occurs within 100 years is 0.65, with 95% credibility intervals of 0.43 and 0.84. If the time interval for historical records is increased to 150 or decreased to 50 years, the mean posterior predicted probability decreases to 0.53 or increases to 0.80, respectively. This analysis suggests there is a very good chance that catchments may have no recorded debris flow activity over long periods yet pose an unacceptable and unrecognised risk to life from debris flows.

If we use 95% credible intervals for $P_H$ (1/500, 1/200), there is also a very good chance that catchments may have no recorded debris flow activity over long periods. However, in this case the risk to life from debris flows is considerably less (probability that $P_H > P_H(max) = 0.006$).

## 4   Discussion

### 4.1 Uncertainty in parameter values

The model parameters (Eq. 4) determining the $P_H(max)$ were based on reported values in the literature. All have uncertainty, but some appear to have higher uncertainty than others. The probability of impact on dwellings if a debris flow occurs ($P_{S:H}$) is assumed to be 1. Where the fan is small and/or dwellings are sited in the likely path for a debris flow, this is a credible assumption. If dwellings are sited at a distance from the flow path, it is a matter of whether the debris flow avulses, and if it does, will it travel towards dwellings sited on the fan? Debris flow avulsion is poorly understood, and patterns of deposition on debris-flow fans have been monitored or reconstructed on only a few natural debris-flow fans (e.g. Zubrycky et al., 2021; de Haas et al., 2018; Santi et al., 2017).

The probabilities of an individual death if dwelling impact occurs ($V$) and that an individual will be present when the landslide occurs ($P_{T:S}$) are even more uncertain, depending on the interaction of debris flow intensity, dwelling vulnerability and human behaviour. The temporal probability that an individual will be present depends on human behaviours such as evacuation, sheltering in place, diurnal variations in occupancy, or seasonal variations in occupancy, as found with holiday homes. In New Zealand, given the large number of debris flow impacts on dwellings within the last 15 years with no fatalities (albeit with injuries and lucky escapes), the values for $V$ (0.1) and $P_{T:S}$ (0.69) may be too high. However, given the risk-to-life implications of these parameters, we have adopted a precautionary approach.

Finally, the model must deal with catchments where there is not enough information to estimate ARIs for life-threatening debris flows. Based on estimates of ARI for life-threatening debris flows from four New Zealand studies, we used 95% credible intervals for $P_H$ of (1/500,1/100) and (1/500,1/200) to estimate the probability that $P_H(max)$ would be exceeded for a debris flow catchment. We found that the choice of the lower threshold for the credible interval was critical. If we used 1/200 (ARI=200 years), then the probability was low that the risk-to-life threshold (0.001) would be exceeded. However, if the lower threshold for the credible interval was 1/100 (ARI=100 years), then the probability that the risk to life threshold (0.001) would be exceeded was much higher. Again, a cautious approach would be to assume 95% credible intervals for $P_H$ of (1/500,1/100) and, therefore, a significant risk to life (probability that $P_H > P_H(max) = 0.2560$).

## 4.2 Limitations of the model

Our model assumes a single annual probability threshold for a debris flow that is an unacceptable risk to life for occupiers of a dwelling in a debris flow catchment rather than a more complex and realistic model that integrates a range of debris flow frequency and intensity scenarios.

An example of this limitation of our model is the "window of non-recognition" estimate, where catchments may exhibit no debris flow activity over long periods yet pose an unacceptable and unrecognised risk to life from debris flows. Of course, this analysis is based on limited data for ARIs of life-threatening debris flows in four catchments. For catchments with smaller ARIs, the proportion of unrecognised catchments with zero occurrences will be smaller, and that of recognised catchments with occurrences ≥1 will be larger. At the same time, these more frequent debris flows may not be life-threatening, leading to complacency about the actual risk to life in the catchment. This was the case for Matatā township in the eastern Bay of Plenty, New Zealand. Four debris flows had occurred at Matatā since 1860 before a major debris-flow disaster in 2005, which destroyed 27 dwellings and damaged 87 properties, fortuitously with no fatalities (McSaveney et al., 2005).

Despite these limitations, we have chosen a simple model because reliable data are scarce, and most model parameters are subject to considerable uncertainty. More importantly, our conceptual approach highlights the dangers of complacency about the risk to life from debris flows. Using simple concepts and Bayesian inference, we can show that, given precautionary but realistic assumptions about debris flow hazards and the vulnerability of dwellings and

their occupants, unrecognised catchments with no history of debris flow activity can pose an unacceptable risk to life. Parameters subject to uncertainty (debris flow ARIs, probability of debris flow impact, dwelling vulnerability, occupancy during debris-flow triggering rainfall events) must be priorities for research to better understand the risk to life from debris flows.

## 5    Conclusions

Debris flows are a potentially dangerous natural hazard for any dwelling on an alluvial fan at the mouth of a steepland catchment. However, debris flow-susceptible catchments may be unrecognised because debris flows may only rarely occur in each catchment. Even where reconnaissance studies using morphometric indices (e.g., Melton $R$) indicate a significant potential hazard, the long annual recurrence intervals (ARIs) for some debris flow catchments mean their annual probability of occurrence ($P_H$) is difficult to estimate reliably. Thus, there is a danger that their risk may be considered negligible.

Here, we have handled this difficulty by inverting the problem. Instead of trying to estimate $P_H$ for debris flows in a specific catchment, we have back-calculated a maximum acceptable annual probability $P_H(\text{max})$ to meet accepted thresholds for maximum risk to life. This has allowed us to:

1. Compare the threshold $P_H(\text{max})$ with four New Zealand studies where the probability distribution of $P_H$ can be estimated from field evidence. Given conservative assumptions about the debris flow ARI, probability of impact on dwellings and the probability of mortality for an impacted dwelling, we have shown that for catchments with one dwelling, $P_H$ can exceed $P_H(\text{max})$.

2. Estimate the "window of non-recognition" where debris flows within a catchment may be so infrequent that it is not recognised as susceptible, yet the risk to life from debris flows exceeds the accepted threshold. We have shown that a significant proportion of debris-flow-susceptible catchments will fall within this window, even assuming up to 150 years of written or oral history recording debris flows within the catchment.

3. Explore the influence of the important parameters underlying the annual risk to life from debris flows. The observed frequency of deaths in New Zealand dwellings from debris flow impacts, admittedly from a small sample, appears to be lower than the assumed value in this study, suggesting that these key parameters need further research.

4. Nevertheless, we have shown that catchments not recognised as debris-flow-capable
        can pose risks to life that are unacceptable. Land-use planning for future
        developments in a potentially susceptible catchment cannot rely on the fact that no
        debris flows have been recorded. There is a need to do site analyses and to think
        carefully about the siting of dwellings or other structures that people may occupy.

*Author contributions.* MB and TD conceptualized and designed the study, including a
literature review and preliminary analyses. EM performed statistical analyses, produced
figures, and contributed to the writing and reviewing of the manuscript. TR and DP
contributed to the writing and reviewing of the manuscript. MB revised and edited the text.

*Competing interests:*
The authors have no competing interests.
*Funding and/or conflicts of interests/competing interests.* None of the authors has any
financial or non-financial interests directly or indirectly related to or in conflict with the work
submitted for publication.
*Acknowledgements.* This paper arose from our work on mapping catchment debris flow
susceptibility at a regional scale for Marlborough and Tasman District Councils in New
Zealand. We acknowledge the support of those two councils and the Envirolink scheme,
which funds research organisations to provide councils with advice and support for research.

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
