# Peer review of "Identifying unrecognised risks to life from debris flows"

_EGUsphere, 2023_

## Referee Comment (RC2)

**Paper Review: Exploratory analysis of the annual risk to life from debris flows**

*Bloomberg et al., 2023*

The paper presents a unique and interesting way of tackling the challenge of lack of information on hazard frequency and the implications for risk. The authors outline a methodological framework, using a Bayesian approach, to identify "windows of non-recognition" where debris flows return periods are enough to have not occurred in the recent historical record, but still may result in an intolerable risk to the population that have settled on the fan. This methodology has potential to be very useful for hazard and risk management. The paper is clearly written and easy to follow.

However, the manuscript as is requires major modifications, including more evidence to underpin some of the parameter assumptions as well as greater clarity about the purpose and key message of the manuscript. Further sensitivity analysis of the other key risk calculation parameters is needed to back up bullet point 4 of the conclusion. This sensitivity analysis may add to the strength of the argument for identifying the "window of non-recognition". Throughout the manuscript there needs to be clearer links to the wider debris flow and risk literature. Re-structuring the introduction, discussion and conclusion may more clearly focus on the key message around the methodological framework and end purpose of the information.

There needs to be more links to the literature on risk evaluation and acceptable and tolerable risk thresholds. All the associated risk definitions need to have much tighter definitions. For example, it would be good to define what is included in unacceptable risk (i.e., intolerable and tolerable risk or just intolerable risk), as this was unclear to me throughout the paper.  Unacceptable is term not  often used or recommended given the two possible meanings stated above.

The same is needed for the calculations and definitions of individual and societal risk. Additionally, there is not a linear relationship between individual and societal risk. Societal risk calculation methods should follow those presented in Strouth and McDougall, 2020 and are scenario based. Is the method intended to calculate the Annual Probable Lives Lost (APPL) from the PH max event? And use APPL to back calculate PHmax. Given that you are evaluating the probability of only one debris flow scenario occurring (PHmax) this calculation route may be okay but requires further checking and a more in depth and robust explanation.

How does your stated risk threshold (which is based on individual risk) relate to societal/group risk, as these risk thresholds are often developed separately? Can this be explored in more depth? Linked to this is providing justification for considering both individual and societal risk, rather than just one or the other. How may this influence risk managers decision-making?

I was unconvinced by the assumption of the probability of spatial impact calculation. Is it possible to include within the sensitivity analysis an evaluation of this term, as it will likely have a big impact on the risk value. Can prior research such as Zubrycky et al 2021 provide distributions to evaluate within your Bayesian framework? This needs further links to the literature and explored more in the discussion. Is the Zubrycky et al., 2021 approach something that could be adopted in NZ?

Additionally, P S:H is a conditional probability of the hazard event occurring and therefore the equation 3 and 4 may need to be re-written to account for this conditional probability (e.g., 1 – (1-Prob) where prob is the individual hazard conditional probability).

**Specific Comments**

Line 35: Reference for "Debris flows as an unrecognised and underappreciated hazard…" Why is that the case?

Line 39: Remove the colloquial term "landslips"

Section 2.1: Move information about need for methodological framework to introduction.

Line 107: change from "is necessary" to "may be necessary"

Line 165: Missing references to paragraph

Line 189: Feels like this belongs in the discussion?

Table 1: Doesn't match earlier description in text. Would be good to provide a separate overview table of published debris flow case-studies and associated ARI in NZ

Line 330: Missing references for this section. Pollock and Wartman 2020 – Human vulnerability to landslides might be useful here.

Line 335: Is this because we can't always capture dynamic risk parameters (e.g., exposure and evacuation with heavy rainfall)? It would be good to highlight the need for dynamic risk models.

**Literature mentioned in comments:**

Strouth and McDougall, 2022 – Individual risk evaluation for landslides: key details

Zubrycky et al., 2021 – Exploring new methods to analyse spatial impact distributions on debris-flow fans using data from south-western British Columbia

Sim et al., 2022 – A review of landslide acceptable and tolerable risk

Pollock and Wartman, 2020 – Human vulnerability to landslides

S de Vilder

---

## Author Comment (AC1)

| Reviewer 1 | | |
|---|---|---|
| **Line** | **Reviewer comment** | **Response** |
| | | We thank the reviewer for their comments. We emphasise that our paper proposes a way to estimate risk to life in the absence of any information apart from regional-scale morphometric analysis such as the Melton ratio. Many of the reviewer's comments regarding our assumptions are correct—but note that our aim is to demonstrate the possibility of risk to life, where no risk is perceived by communities or their decision-makers. To do this easily and cheaply on a regional scale, we need to make "precautionary but realistic" assumptions. These assumptions therefore will err on the side of caution. If our analysis gains the attention of communities and their decision-makers, we are able to investigate in more detail. |
| | However, this key finding is currently vague, partly because it is difficult to estimate risk parameters and partly because the current manuscript has limited/unclear presentation of the specific return intervals that result in unacceptable risk. Are they intended to be global/generalizable estimates, or regionally specific? Fully characterizing hazard and risk for decision-making would require additional process-based investigation or empirical approaches that use local/regional data. I encourage the authors to describe how their model results could be used to support actionable strategies for prioritizing further risk-reduction efforts. | Our purpose is simply to demonstrate the need for risk reduction—prioritisation would be the next step. At least in NZ, but we suspect elsewhere, the risk to life and property from debris flows is often ignored by communities and their decision-makers. So even to get acknowledgement of the potential for a problem is an achievement—it is not a trivial task. We will bring this point to the fore in any revision. |

| Line | Reviewer comment | Response |
|---|---|---|
| | Significant re-framing to better reflect the conclusions. This could involve a more specific paper title, updates to the abstract, introduction, and conclusions that better reflect the contributions of their work. Specifically, I would suggest that the authors re-align the work with the primary contribution as described in the first paragraph of this review. | Agreed—see our comments above. |
| | Deeper grounding in real-world processes. This could be accomplished through improved basis in the literature and in consideration of both physical and social processes. Most importantly, the authors mention field evidence in the conclusions, but do not introduce study areas, present field observation, or describe data collection methodologies. These are critical for understanding the scientific contribution as well as its ability to be generalized or extrapolated to other study regions. | Our paper is a "methods" paper, and describes a straightforward method based on well-accepted principles. If one accepts the underlying principles (which are widely accepted in landside risk management) and understands that the assumed variable values are based on best available published data, it has the potential to be a valuable tool to create awareness of debris flow risks where that awareness is lacking. We do not see it as a novel contribution in scientific terms—far from it. But we believe it is a novel way to communicate debris flow risks when so many communities are oblivious or complacent. We ask that it be judged according to that criterion. |
| | Expanded/restructured introduction and methods section that present existing knowledge and the research question (introduction) and the author's novel approach (methods). | |
| | Finally, the authors describe a single, universal "window of non-recognition," which reflects the generic or best estimate parameter values for risk. However, considering that risk estimates and the window of observation relies on settlement periods, risk tolerance, and physical processes (probability of avulsion), I am not convinced that this value would apply to broad areas. Instead, I suggest that the authors present their window with a methodology framework, investigate a specific study area, and/or evaluate a wider range of parameter values for each risk parameter. | Thank you for this suggestion. However, we do ask the reviewer to reconsider this point. Our methodology can be applied to any situation where there are potential risks from debris flow hazards. However, the nature of the risks and their assessments will vary widely, depending on the factors mentioned by the reviewer (settlement periods, risk tolerance, and physical processes such as probability of avulsion). We do show how these can be included in the indicative assessment of risk to life. We have used a case study, using ARIs from two well-known life-threatening debris flow events in NZ. |

| Line | Reviewer comment | Response |
|---|---|---|
| | A few specific recommendations are described in the line comments, but in general I recommend separating background contextual information in the introduction, with specific methodology descriptions for the author's work in the methods. Currently, the methods section describes the research problem, which would be better suited in an introduction section. | Agreed. |
| 1 | Consider a more descriptive title which highlights specific analysis or finding of your work. | Agreed, we could change the title to "Identifying unrecognised risk to life from debris flows" |
| 24 | "No history of debris flows," is too vague. Do you mean geologic history? Oral history? Written history? Consider describing your study area, as settlement history is also highly variable around the world. Expanding urban areas and limited records absolutely result in low public awareness of debris flow hazards, but consider acknowledging that human settlement in New Zealand is relatively short (hence the challenge the authors' research question) densely inhabited areas in Eurasia may have hundreds of years of detailed written records, and oral histories for many indigenous peoples describe landslides or other debris flows over millennia. | A fair point. "No history of debris flows" could be rewritten as "no current knowledge about previous occurrence of debris flows". This current knowledge could be obtained from many different sources, the key thing is that the community and its decision makers are currently unaware of any hazard. Agreed, awareness of debris flows is variable around the world, but in many places (not just NZ) it is low. Sure, a long history of settlement may mean that communities have written or oral records dating back centuries or more. But we still see landslide disasters in long-settled regions such as China or South America. In many cases, rapid population increases and/or poverty have forced people to settle in areas that were previously not occupied. We do recognise NZs short human settlement history in the text (line 200 et seq.) |
| 28-34 | Please provide citations to the literature so that readers can seek additional context on debris flow processes, sediment pathways, and debris flow hazards in New Zealand. | Agreed, this would be useful, and a selection of papers can be cited. |

| Line | Reviewer comment | Response |
|---|---|---|
| Section 2.1 | Section 2.1. Much of this section would be better suited to the introduction. | Agreed. |
| 58 | Table 1 does not summarize ARIs for different catchments. Please add the summary table and revise your in-text citation. | We ask the reviewer to reconsider this point. For most catchments, we do not have ARIs. We have a few reports which estimate ARIs for several well-known life-threatening debris flow catchments in NZ, that is all. Instead, we invert the problem and say that if we assume a plausible ARI (200-500 years) does this result in an unacceptable risk to life? We use a threshold of 0.001 for annual RTL, but discussion in the NZ Geotechnical Society (2023) describes annual individual fatality rates of 0.0001--our analysis can easily accommodate different choices of threshold for RTL. |
| 62 | It is true that field evidence & topographic analysis can be costly to collect and process. However, I would argue that these are the best tools for developing specific hazards understanding & precise risk estimates. Your approach for risk assessment should not replace process-based assessment, but may be useful in prioritizing communities/residences for improved outreach & risk awareness. | We certainly have no intention of supplanting the need for field work, modelling etc. Our purpose is simply to demonstrate the need for risk reduction—prioritisation would be the next step. At least in NZ, but we suspect elsewhere, the risk to life and property from debris flows is ignored by communities and their decision-makers. So even to get acknowledgement of the potential for a problem is an achievement—it is not a trivial task. |
| 68-70 | Consider "catchment gradient is associated with debris flow occurrence," and reference other specific topographically based tools for landslide susceptibility (E.g., Montgomery et al., 1994; Dietrich et al., 2001). | Agreed, there are a broad range of methods for topographically based tools for assessing debris flow susceptibility, on both a regional and site-specific basis. A brief review of these could be included in the paper. |
| 70-71 | That identifies catchments likely to produce debris flows, which is easily calculated for many catchments over large areas, even where topographic resolution is poor or computation is limited. Also it sounds like these values have already been calculated for large areas of NZ). | The authors have completed several regional-scale investigations using Melton-R as a metric, as well as testing other methods e.g. Flow-R. However, these methods cover only limited areas of debris flow susceptible terrain in NZ. |

| Line | Reviewer comment | Response |
|---|---|---|
| 96-97 | I understand that you need to determine threshold risk values, but consider providing more context on how and why risk tolerance may vary, and why these values ($10^{-3}$-$10^{-4}$) are appropriate according to Taig. et al. Here or in the discussion, you may want to acknowledge that "unacceptable" risk reflects the values & tolerances of individuals and communities. | Agreed, we will revise this text to include an extended justification for choice of these threshold risk values. |
| Eqn 2 | Are these standard abbreviations? I found them hard to follow (where does the "H" come from? Is PS:H a ratio? E stands for exposure?) Consider using simpler abbreviations or adding some description to help readers keep track of which parameter is which. | These variable names are the same or similar to the cited literature on risk to life e.g. Walker et al., 2007; Porter and Morgenstern, 2012; de Vilder et al., 2022. Since PH is the probability of a debris flow event, PS:H is the spatial (S) probability of an impact, given that the debris flow has occurred (H). We'd like to stick with this notation to ensure consistency with other published papers. |
| 132-134/147 | I'm not sure I agree with this estimate. In my experience, very few debris flows impact the entire debris flow fan. While it is challenging to estimate the probability of avulsion, are there any estimates in the literature which might describe the distribution of areas, as a proportion of total fan area, that occur during a debris flow? De Haas and others, or works by C Scheidl, DM Staley, or D Rickenmann may be useful places to look for an estimate. | It is a fair criticism. However, there is no simple, easily applied solution to this problem. We think it is more pragmatic to take an upper-bound approach here. |
| 165-167 | It may be worth adding a section in the discussion on how further investigation could be used to refine generic estimates. Sediment volume calculations, for example, could be used to improve estimates of debris flow area and deposit depths for exposure calculations. Consider expanding on this statement and adding appropriate support from the literature. | OK, but for our level of analysis we would only seek to refine estimates of variable values to a limited extent. We believe the sequence is--demonstrate the possibility of a risk to life (and/or property). Then prioritise, using regional-scale mapping of debris flow susceptibility and potential assets at risk. Would not assessment of potential debris flow severity and exposure of assets would be best focussed on studies at the site or community level? |

| Line | Reviewer comment | Response |
|---|---|---|
| 183-184 | These values are lower than I would expect. Can you provide more detail about the data used to make these estimates? Considering that your estimates are intended to describe worst-case scenarios where whole debris flow fans are inundated, 0.1 seems far too low. | The data were published values, so we do not have access to the data. We do not explicitly aim to model "worst-case" scenarios with complete inundation of fans. These were just credible values from the literature, and we used them to demonstrate how our model works. As we note in the text, it would be desirable to come up with variable values that were estimated or calibrated against observational data. |
| 208 | I don't agree with this assumption, and assuming the worst-case scenario conflicts with your goal of best-estimate risk calculation. | Extension of our model from individual risk to life to a "societal" risk of multiple deaths is a weakness in our model. Extending our analysis from individual risk to life to risk of multiple deaths requires knowledge of the variation amongst individuals in terms of risk variables e.g. PS:H and V. We do not have this knowledge-- the best we might be able to do is assume uniform values for all individuals (but see comment re PS:H below) |
| Figure 1 | Figure 1. What is the value shown on the y-axis? | This is a relative likelihood that that the value of the x-variable (PH) would be equal to that PH value of a random sample for the population. It is a relative likelihood, so the actual values on the axis are not so important. They could be normalised i.e. make the y-axis take values that will make the total area under the histogram equal to 1, which is saying that the total probability of the entire distribution of events is equal to a probability of 1. This would be more intuitive. |
| Figure 2 | Figure 2. Can you provide a clearer description of the populations shown in this figure? Are these real-world catchments or monte carlo-type simulations? | See lines 230 et seq. The frequency distributions in Figs 1 and 2 are not generated from Monte-Carlo simulations. Instead, we assume a specific type of frequency distribution, the beta distribution. This has two parameters, which were chosen to correspond to a population where 95% of the population occurred within 1/500 and 1/200 for the parameter PH (annual probability of a debris flow occurrence). |

| Line | Reviewer comment | Response |
|---|---|---|
| 390 | No field evidence is presented here. Please add description and summary of field evidence that you use to draw conclusions. | Agreed—the way this statement is made implies that we analysed data.  The comment was more of an exploratory nature—deaths from debris flows in NZ are low, therefore if we are showing that RTL may be unacceptable in debris flow catchments with settlement, we need to reconcile these two pieces of information. We will rewrite to show that our comment is intended to raise an issue, rather than resolve it. |

---

## Author Comment (AC2)

| Reviewer 2 | | |
|---|---|---|
| Line | Reviewer comment | Response |
| | Requires major modifications, including more evidence to underpin some of the parameter assumptions as well as greater clarity about the purpose and key message of the manuscript. | We thank the reviewer for their comments. We emphasise that our paper proposes a way to demonstrate the possibility of risk to life in the absence of any information apart from regional-scale morphometric analysis, such as the Melton ratio. Many of the reviewer's comments regarding our assumptions are correct--but note that we aim to demonstrate the possibility of risk to life, where communities or their decision-makers perceive no risk. To do this easily and cheaply on a regional scale, we need to make "precautionary but realistic" assumptions. These assumptions therefore will err on the side of caution. If our analysis gains the attention of communities and their decision-makers, we are in a position to investigate in more detail. " |
| | Further sensitivity analysis of the other key risk calculation parameters is needed to back up bullet point 4 of the conclusion. This sensitivity analysis may add to the strength of the argument for identifying the "window of non-recognition | Agreed--this conclusion is not sufficiently supported by the preceding text. It is not a critical part of our argument; the simplest option is to delete it. |
| | Throughout the manuscript, there needs to be clearer links to the wider debris flow and risk literature. Re-structuring the introduction, discussion and conclusion may more clearly focus on the key message around the methodological framework and end purpose of the information. | Agreed |

| Line | Reviewer comment | Response |
|---|---|---|
| | How does your stated risk threshold (which is based on individual risk) relate to societal/group risk, as these risk thresholds are often developed separately? Can this be explored in more depth? Linked to this is providing justification for considering both individual and societal risk, rather than just one or the other. How may this influence risk managers decision-making? | Here is our understanding. From Strouth and McDougall(2020). "**Individual risk** is the probability that a specific individual will be killed by a landslide. This risk is often assessed for the individual most at risk within a landslide hazard zone or building and is expressed as the probability of death to an individual (PDI). Strouth and McDougall point out **that** societal/group risk is a more complex concept, but "in practice, at least for landslide risk management decisions in Western Canada, societal risk refers more narrowly to the relationship between the probability of, and number of, people killed."   We used this definition of societal risk. On a specific fan impacted by a debris flow, the number of deaths will depend on the number of people who occupy that fan, which dwellings are impacted, whether individuals are present, and their vulnerabilities. The reviewers are correct; extending our analysis from individual risk to life to risk of multiple deaths requires knowledge of the variation amongst individuals in terms of these risk variables. We do not have this knowledge--the best we might be able to do is assume uniform values for all individuals (but see comment re PS:H below) |
| | I was unconvinced by the assumption of the probability of spatial impact calculation. Is it possible to include within the sensitivity analysis an evaluation of this term, as it will likely have a big impact on the risk value. Can prior research such as Zubrycky et al 2021 provide distributions to evaluate within your Bayesian framework? This needs further links to the literature and explored more in the discussion. Is the Zubrycky et al., 2021 approach something that could be adopted in NZ? | To answer the easiest question—the research by Zubrycky et al. looks useful and relevant to NZ and likely many parts of the world. To answer the more difficult question--could the approach by Zubrycky et al be included in our framework? We argue that to do so negates the aim of our approach.   We aim to demonstrate the possibility of risk to life **exists** where communities or their decision-makers perceive no risk. To do this easily and cheaply on a regional scale, we need to make "precautionary but realistic" assumptions. These assumptions therefore will err on the side of caution. If our analysis gains the attention of communities and their decision-makers, we are in a position to investigate in more detail. As we note, "The observed frequency of deaths in New Zealand dwellings from debris flow impacts, although admittedly a very small sample, appears to be lower than the assumed value in this study, suggesting that these key parameters (in our model) need further research." We think the research by Zubrycky et al. is a promising approach for more detailed investigations and modelling. |

| Line | Reviewer comment | Response |
|---|---|---|
| 35 | Reference for "Debris flows as an unrecognised and underappreciated hazard…" Why is that the case? | This statement is from McSaveney et al., 2005. We suggest that this lack of recognition by the public is partly due to confusing terminology, with previous events referred to as "floods", "flash floods", or "slips" (McSaveney et al., 2005). However, awareness has grown in the NZ natural hazard community, and it is not fair or accurate to say that, currently, there is no awareness. NZ natural hazard scientists and practitioners are increasingly aware of the hazards posed by debris flows. The problem of public and political unawareness remains. Apart from the problem re terminology in media reporting, the other main reason is identified in the paper: "the long ARIs for these events create an illusory sense of security so that their risk to life is not recognised" and our paper addresses this. |
| 39 | Line 39: Remove the colloquial term "landslips" | Agreed, in NZ, "shallow landslides" is the commonly used terminology and we will use this. |
| Section 2.1 | Section 2.1: Move information about need for methodological framework to introduction. | Agreed. |
| 107 | Change from "is necessary" to "may be necessary" | Agreed. |
| 165 | Missing references to paragraph | |
| 189 | Feels like this belongs in the discussion? | Agreed--will revise. |
| Table 1 | Table 1: Doesn't match earlier description in text. Would be good to provide a separate overview table of published debris flow case-studies and associated ARI in NZ | Apologies—but we have carefully reviewed the variables in Table 1 and they match the variables and their descriptions in Section 2.2. Can the reviewer give us a more specific idea of where Table 1 does not cross-reference to the earlier text? Secondly, we are only aware of estimated ARIs for three case studies in NZ--Matata, Thames and Ligar Bay. Matata and Ligar Bay are cited in the text. We could add the Thames estimate (ARI~500 years). This reinforces the point about a lack of information and awareness, certainly in NZ. |
| 335 | Is this because we can't always capture dynamic risk parameters (e.g., exposure and evacuation with heavy rainfall)? It would be good to highlight the need for dynamic risk models. | Agreed, and also the need to collect data to calibrate and test these models. |

---

## Author Response (AR1)

| Reviewer 1 | | |
|---|---|---|
| Line | Reviewer comment | Response |
| | | We thank the reviewer for their comments. We emphasise that our paper proposes a way to estimate risk to life in the absence of any information apart from regional-scale morphometric analysis such as the Melton ratio. Many of the reviewer's comments regarding our assumptions are correct—but note that our aim is to demonstrate the possibility of risk to life, where no risk is perceived by communities or their decision-makers. To do this easily and cheaply on a regional scale, we need to make "precautionary but realistic" assumptions. These assumptions therefore will err on the side of caution. If our analysis gains the attention of communities and their decision-makers, we are able to investigate in more detail. |
| | However, this key finding is currently vague, partly because it is difficult to estimate risk parameters and partly because the current manuscript has limited/unclear presentation of the specific return intervals that result in unacceptable risk. Are they intended to be global/generalizable estimates, or regionally specific? Fully characterizing hazard and risk for decision-making would require additional process-based investigation or empirical approaches that use local/regional data. I encourage the authors to describe how their model results could be used to support actionable strategies for prioritizing further risk-reduction efforts. | Our purpose is simply to demonstrate the need for risk reduction—prioritisation would be the next step. At least in NZ, but we suspect elsewhere, the risk to life and property from debris flows is often ignored by communities and their decision-makers. So even to get acknowledgement of the potential for a problem is an achievement—it is not a trivial task. We bring this point to the fore in our revised text, adding the following text: *There is a large and growing literature on debris-flow hazard assessments (Jakob, 2021), but these assessments typically required funding in order for them to be made. Thus, the debris flow literature has an inherent bias towards relatively complex studies involving a range of site assessment and modelling techniques. There is a lack of studies that describe how to overcome the problem described by Jakob (2021): "Most districts, states, provinces, or even nations have limited funds for geohazard mitigation. This necessitates the allocation of existing funds to those sites with the highest risk potential. Funds for studies and mitigation often get allocated because of particularly damaging events that result in focused public, media, and political attention. Those sites, however, may not necessarily be the ones with highest risk."* |

| | Significant re-framing to better reflect the conclusions. This could involve a more specific paper title, updates to the abstract, introduction, and conclusions that better reflect the contributions of their work. Specifically, I would suggest that the authors re-align the work with the primary contribution as described in the first paragraph of this review. | Agreed—see our comments above.  Note that we have revised the title to better reflect the purpose of the paper |
|---|---|---|
| | Deeper grounding in real-world processes. This could be accomplished through improved basis in the literature and in consideration of both physical and social processes. Most importantly, the authors mention field evidence in the conclusions, but do not introduce study areas, present field observation, or describe data collection methodologies. These are critical for understanding the scientific contribution as well as its ability to be generalized or extrapolated to other study regions | Our paper is a "methods" paper, and describes a straightforward method based on well-accepted principles. If one accepts the underlying principles (which are widely accepted in landside risk management) and understands that the assumed variable values are based on best available published data, it has the potential to be a valuable tool to create awareness of debris flow risks where that awareness is lacking.
We do not see it as a novel contribution in scientific terms—far from it.  But we believe it is a novel way to communicate debris flow risks when so many communities are oblivious or complacent. |
| | Expanded/restructured introduction and methods section that present existing knowledge and the research question (introduction) and the author's novel approach (methods). |  We have substantially revised the text in line with the reviewer's suggestions. |

| | | Finally, the authors describe a single, universal "window of non-recognition," which reflects the generic or best estimate parameter values for risk. However, considering that risk estimates and the window of observation relies on settlement periods, risk tolerance, and physical processes (probability of avulsion), I am not convinced that this value would apply to broad areas. Instead, I suggest that the authors present their window with a methodology framework, investigate a specific study area, and/or evaluate a wider range of parameter values for each risk parameter. | Thank you for this suggestion.  However, we do ask the reviewer to reconsider this point.  Our methodology can be applied to any situation where there are potential risks from debris flow hazards.  However, the nature of the risks and their assessments will vary widely, depending on the factors mentioned by the reviewer (settlement periods, risk tolerance, and physical processes such as probability of avulsion).

We do show how these can be included in the indicative assessment of risk to life.  We have used a case study, using ARIs from four well-known life-threatening debris flow events in NZ. |
| | | A few specific recommendations are described in the line comments, but in general I recommend separating background contextual information in the introduction, with specific methodology descriptions for the author's work in the methods. Currently, the methods section describes the research problem, which would be better suited in an introduction section. | Agreed.  We have reorganised the text so that the research problem is stated in a separate section from the Methods. |
| | 1 | Consider a more descriptive title which highlights specific analysis or finding of your work. | Agreed, we could change the title to "Identifying unrecognised risk to life from debris flows" |

| 24 | "No history of debris flows," is too vague. Do you mean geologic history? Oral history? Written history? Consider describing your study area, as settlement history is also highly variable around the world. Expanding urban areas and limited records absolutely result in low public awareness of debris flow hazards, but consider acknowledging that human settlement in New Zealand is relatively short (hence the challenge the authors' research question) densely inhabited areas in Eurasia may have hundreds of years of detailed written records, and oral histories for many indigenous peoples describe landslides or other debris flows over millennia. | A fair point. "No history of debris flows," could be rewritten as " "No recorded history of debris flows". This recorded knowledge could be obtained from many different sources, the key thing is whether the community and its decision makers are currently unaware of any hazard. Agreed, awareness of debris flows is variable around the world, but in many places (not just NZ) it is low. Sure, a long history of settlement may mean that communities have written or oral records dating back centuries or more. But we still see landslide disasters in long-settled regions such as China or South America. In many cases, rapid population increases and/or poverty have forced people to settle in areas that were previously not occupied. We do recognise NZs short human settlement history in the text. |
|---|---|---|
| 28-34 | Please provide citations to the literature so that readers can seek additional context on debris flow processes, sediment pathways, and debris flow hazards in New Zealand. | We cite the following:
Beca Ltd: Natural hazards affecting Gorge Road, Queenstown. Prepared for Queenstown Lakes District Council, Beca Ltd, Christchurch, New Zealand, 2020.
Bloomberg, M. and Palmer, D.J.: Estimation of catchment susceptibility to debris flows and debris floods–Marlborough Sounds, Pelorus Catchment and Wairau Northbank. Draft Report to Marlborough District Council,
https://www.marlborough.govt.nz/repository/libraries/id:2ifzri1o01cxbymxkvwz/hierarchy/documents/your-council/meetings/2022/environment-2022/Item_5-17032022-Estimation_of_catchment_susceptibility_to_debris_flows.pdf, 2022.
Farrell J. and Davies T.: Debris flow risk management in practice: a New Zealand case study, Association of Environmental and Engineering Geologists; Special Publication 28. 2019.
Massey, C.I., Thomas, K-L., King A.B., Singeisen, C., Horspool, N.A. and Taig, T. SLIDE (Wellington): vulnerability of dwellings to landslides (Project No. 16/SP740), GNS Science report; 2018/27), GNS Science, Lower Hutt, New Zealand, 2018. |

| | | |
|---|---|---|
| | | McSaveney, M., Beetham, R., and Leonard, G.: The 18 May 2005 debris flow disaster at Matata: Causes and mitigation suggestions, GNS Science Client Report, 2005/71. GNS Science, Wellington, New Zealand, 2005.
McSaveney, M. and Beetham, R.: The potential for debris flows from Karaka Stream, Thames, Coromandel, GNS Science Consultancy Report, 2006/014, GNS Science, Wellington, New Zealand, 2006.
Page, M., Langridge, R., Stevens, G., and Jones, K.: The December 2011 debris flows in the Pohara-Ligar Bay area, Golden Bay: causes, distribution, future risks and mitigation options, GNS Science Consultancy Report 2012/305, GNS Science, Wellington, New Zealand, 2012.
Welsh, A. and Davies, T.: Identification of alluvial fans susceptible to debris-flow hazards, Landslides, 8, 183–194, 2011. |
| Section 2.1 | Section 2.1. Much of this section would be better suited to the introduction. | Agreed. We have rearranged the text accordingly. |
| 58 | Table 1 does not summarize ARIs for different catchments. Please add the summary table and revise your in-text citation. | We ask the reviewer to reconsider this point.  For most catchments, we do not have ARIs.  We have a few reports which estimate ARIs for several well-known life-threatening debris flow catchments in NZ, that is all.  Instead, we invert the problem and say that if we assume a plausible ARI (100-500 years) does this result in an unacceptable risk to life?  We use a threshold of 0.001 for annual RTL, but discussion in the NZ Geotechnical Society (2023) describes annual individual fatality rates of 0.0001--our analysis can easily accommodate different choices of threshold for RTL.
Note that we do list the data for the four NZ catchments where we have expert assessment of ARIs. |
| 62 | It is true that field evidence & topographic analysis can be costly to collect and process. However, I would argue that these are the best tools for developing specific hazards understanding & precise risk estimates. Your approach for risk assessment should not replace process-based assessment, but may be useful in prioritizing communities/residences for improved outreach & risk awareness. | We certainly have no intention of supplanting the need for field work, modelling etc.  Our purpose is simply to demonstrate the need for risk reduction—prioritisation would be the next step.  At least in NZ, but we suspect elsewhere, the risk to life and property from debris flows is ignored by communities and their decision-makers.  So even to get acknowledgement of the potential for a problem is an achievement—it is not a trivial task.  We refer again to the quote from Jakob (2021) in the text from the paper. |

| | | |
|---|---|---|
| 68-70 | Consider "catchment gradient is associated with debris flow occurrence," and reference other specific topographically based tools for landslide susceptibility (e.g., Montgomery et al., 1994; Dietrich et al., 2001). | Agreed, there are a broad range of methods for topographically based tools for assessing debris flow susceptibility, on both a regional and site-specific basis.  A brief review of these has been included in the revised paper.
However, we do not want to cover this point in depth.  Our intention was to define the problem i.e. we can use geospatial methods to locate catchments with high debris flow susceptibility—but then, so what?  Unless we can demonstrate unacceptable risk to life (or property) then communities and their governance organisations will not be motivated to investigate further.  We believe we have made this point clear in the text of the paper. |
| 70-71 | That identifies catchments likely to produce debris flows, which is easily calculated for many catchments over large areas, even where topographic resolution is poor or computation is limited. Also it sounds like these values have already been calculated for large areas of NZ). | The authors have completed several regional-scale investigations using Melton's R as a metric, as well as testing other methods e.g.  Flow-R.  However, these methods cover only limited areas of debris flow susceptible terrain in NZ see Bloomberg and D.J (2022), Welsh and Davies (2011). |
| 96-97 | I understand that you need to determine threshold risk values, but consider providing more context on how and why risk tolerance may vary, and why these values ($10^{-3}$-$10^{-4}$) are appropriate according to Taig. et al. Here or in the discussion, you may want to acknowledge that "unacceptable" risk reflects the values & tolerances of individuals and communities. | We are reluctant to do this.  Our objective was to describe simple methods for assessing debris flow risk to life according to a specific threshold.  Our model can easily accommodate any change to this threshold—but discussion of an appropriate threshold is a big topic and is well covered by other published papers.  We chose an individual threshold of 0.001 as this is commonly used in the literature. |
| Eqn 2 | Are these standard abbreviations? I found them hard to follow (where does the "H" come from? Is PS:H a ratio? E stands for exposure?) Consider using simpler abbreviations or adding some | These variable names are the same or similar to the cited literature on risk to life e.g. Walker et al., 2007; Porter and Morgenstern, 2012; de Vilder et al., 2022.  Since PH is the probability of a debris flow event, PS:H is the spatial (S) probability of an impact, given that the debris flow has occurred (H).  We'd like to stick with this notation to ensure consistency with other published papers. We do explain these variables when they are introduce din the text, and in Table 2. We have adopted the specific version of this notation from Jakob |

| | | |
|---|---|---|
| | description to help readers keep track of which parameter is which. | et al., 2012 to meet the editor's comments on notation.  This may clarify that PS:H is a conditional probability, not a ratio. |
| 132-134/147 | I'm not sure I agree with this estimate. In my experience, very few debris flows impact the entire debris flow fan. While it is challenging to estimate the probability of avulsion, are there any estimates in the literature which might describe the distribution of areas, as a proportion of total fan area, that occur during a debris flow? | Agreed—however our analysis is for the individual or dwelling that is subject to the highest risk to life.  In many cases, especially for smaller catchments (<500ha) associated   with class 3 or 4 debris flows (Jakob, 2005) the dwelling is often located close to the apex of the fan—highest point, best view etc). Our aim is a simple analysis that can be applied to any debris flow catchment.  Considering the effect of avulsion is beyond the scope of our paper. |
| 165-167 | It may be worth adding a section in the discussion on how further investigation could be used to refine generic estimates. Sediment volume calculations, for example, could be used to improve estimates of debris flow area and deposit depths for exposure calculations. Consider expanding on this statement and adding appropriate support from the literature. | OK, but for our level of analysis we would only seek to refine estimates of variable values to a limited extent. We believe the sequence is--demonstrate the possibility of a risk to life (and/or property).  Then prioritise, using regional-scale mapping of debris flow susceptibility and potential assets at risk.  Would not assessment of potential debris flow severity and exposure of assets would be best focussed on studies at the site or community level? |
| 183-184 | These values are lower than I would expect. Can you provide more detail about the data used to make these estimates? Considering that your estimates are intended to describe worst-case scenarios where whole debris flow fans are inundated, 0.1 seems far too low. | The data were published values, so we do not have access to the data. We do not explicitly aim to model "worst-case" scenarios with complete inundation of fans.  These were just credible values from the literature, and we used them to demonstrate how our model works.  As we note in the text, it would be desirable to come up with variable values that were estimated or calibrated against observational data. |

| | | |
|---|---|---|
| 208 | I don't agree with this assumption, and assuming the worst-case scenario conflicts with your goal of best-estimate risk calculation. | Extension of our model from individual risk to life to a "societal" risk of multiple deaths was a weakness in our model. We have attempted to solve this by revising our analysis and limiting it to the individual or dwelling that is subject to the highest risk to life. |
| Figure 1 | Figure 1. What is the value shown on the y-axis? | This is a relative likelihood that that the value of the x-variable (PH) would be equal to that PH value of a random sample for the population.  It is a relative likelihood, so the actual values on the axis are not important. |
| Figure 2 | Figure 2. Can you provide a clearer description of the populations shown in this figure? Are these real-world catchments or monte carlo-type simulations? | The frequency distributions in Figs 1 and 2 are not generated from Monte-Carlo simulations. In Bayesian analysis, the "prior" parameters are based on existing knowledge e.g. expert opinion.  We use three studies of catchments in NZ to set the "priors" based on expert assessment of likely ARIs. We assume a commonly used type of frequency distribution, the beta distribution.  This has two "prior" parameters, which were chosen to correspond to a population where 95% of the population occurred within 1/500 and 1/100 (or 1/200) for the parameter PH (annual probability of a debris flow occurrence).  These parameters can be re-evaluated based on subsequent observation. These are called "posteriors".  In our case, we assumed that no life-threatening debris flows would be observed for 100 years.  This "observation" did not result in big differences between prior and posterior parameters i.e. our assumed ARIs were consistent with zero observed debris flows over a long period (100 years). We have added extra text to explain these points. |
| 390 | No field evidence is presented here. Please add description and summary of field evidence that you use to draw conclusions. | Agreed—the way this statement is made implies that we analysed data.  The comment was more of an exploratory nature—deaths from debris flows in NZ are low, therefore if we are showing that RTL may be unacceptable in debris flow catchments with settlement, we need to reconcile these two pieces of information. We have rewritten the text to show that our comment is intended to raise an issue, rather than resolve it. |

| Reviewer 2 | | |
|---|---|---|
| Line | Reviewer comment | Response |
| | Requires major modifications, including more evidence to underpin some of the parameter assumptions as well as greater clarity about the purpose and key message of the manuscript. | We thank the reviewer for their comments. We emphasise that our paper proposes a way to demonstrate the possibility of risk to life in the absence of any information apart from regional-scale morphometric analysis, such as the Melton ratio. Many of the reviewer's comments regarding our assumptions are correct--but note that we aim to demonstrate the possibility of risk to life, where communities or their decision-makers perceive no risk. To do this easily and cheaply on a regional scale, we need to make "precautionary but realistic" assumptions. These assumptions therefore will err on the side of caution. If our analysis gains the attention of communities and their decision-makers, we are in a position to investigate in more detail. |
| | Further sensitivity analysis of the other key risk calculation parameters is needed to back up bullet point 4 of the conclusion. This sensitivity analysis may add to the strength of the argument for identifying the "window of non-recognition | Agreed--this conclusion is not sufficiently supported by the preceding text. It is not a critical part of our argument; the simplest option is to delete it, which we have done. |
| | Throughout the manuscript, there needs to be clearer links to the wider debris flow and risk literature. Re-structuring the introduction, discussion and conclusion may more clearly focus on the key message around the methodological framework and end purpose of the information. | Agreed.  We have made major changes to the structure of the paper and added many extra references accordingly. |

| | | |
|---|---|---|
| | How does your stated risk threshold (which is based on individual risk) relate to societal/group risk, as these risk thresholds are often developed separately? Can this be explored in more depth? Linked to this is providing justification for considering both individual and societal risk, rather than just one or the other. How may this influence risk managers decision-making? | Here is our understanding. From Strouth and McDougall(2020). "**Individual risk** is the probability that a specific individual will be killed by a landslide. This risk is often assessed for the individual most at risk within a landslide hazard zone or building and is expressed as the probability of death to an individual (PDI). Strouth and McDougall point out that **societal/group risk** is a more complex concept, but "in practice, at least for landslide risk management decisions in Western Canada, societal risk refers more narrowly to the relationship between the probability of, and number of, people killed."   We used this definition of societal risk. On a specific fan impacted by a debris flow, the number of deaths will depend on the number of people who occupy that fan, which dwellings are impacted, whether individuals are present, and their vulnerabilities. The reviewers are correct; extending our analysis from individual risk to life to risk of multiple deaths requires knowledge of the variation amongst individuals in terms of these risk variables. We do not have this knowledge--the best we might be able to do is assume uniform values for all individuals (but see comment re PS:H below)

Therefore we have amended our analysis to only apply for the individual or dwelling that is most at risk for a particular catchment. |
| | I was unconvinced by the assumption of the probability of spatial impact calculation. Is it possible to include within the sensitivity analysis an evaluation of this term, as it will likely have a big impact on the risk value. Can prior research such as Zubrycky et al 2021 provide distributions to evaluate within your Bayesian framework? This needs further links to the literature and explored more in the discussion. Is the Zubrycky et al., 2021 approach something that could be adopted in NZ? | To answer the easiest question—the research by Zubrycky et al. looks useful and relevant to NZ and likely many parts of the world. To answer the more difficult question--could the approach by Zubrycky et al be included in our framework?
We argue that to do so negates the aim of our approach.   Our revised analysis is for the individual or dwelling that is subject to the highest risk to life.  In many cases, especially for smaller catchments (<500ha) associated  with class 3 or 4 debris flows (Jakob, 2005) the dwelling is often located close to the apex of the fan—highest point, best view etc.
Our aim is a simple analysis that can be applied to any debris flow catchment. Considering the effect of avulsion is beyond the scope of our paper. We think the research by Zubrycky et al. is a promising approach for more detailed investigations and modelling and we have referred to this and similar papers in our discussion. |

| | | |
|---|---|---|
| 35 | Reference for "Debris flows as an unrecognised and underappreciated hazard…" Why is that the case? | This statement is from McSaveney et al., 2005. We suggest that this lack of recognition by the public is partly due to confusing terminology, with previous events referred to as "floods", "flash floods", or "slips" (McSaveney et al., 2005). However, awareness has grown in the NZ natural hazard community, and it is not fair or accurate to say that, currently, there is no awareness within that community. NZ natural hazard scientists and practitioners are increasingly aware of the hazards posed by debris flows. We have amended our text accordingly.
The problem of public and political unawareness remains. Apart from the problem re terminology in media reporting, the other main reason is identified in the paper: "the long ARIs for these events create an illusory sense of security so that their risk to life is not recognised" and our paper addresses this. |
| 39 | Line 39: Remove the colloquial term "landslips" | Agreed, in NZ, "shallow landslides" is the commonly used terminology and we will use this. |
| Section 2.1 | Section 2.1: Move information about need for methodological framework to introduction. | Agreed. We have revised the text accordingly. |
| 107 | Change from "is necessary" to "may be necessary" | Agreed. We have revised the text accordingly. |
| 165 | Missing references to paragraph | We have added additional references in this section. |
| 189 | Feels like this belongs in the discussion? | Agreed. We have revised the text accordingly. |
| Table 1 | Table 1: Doesn't match earlier description in text. Would be good to provide a separate overview table of published debris flow case-studies and associated ARI in NZ | Apologies—but we have carefully reviewed the variables in Table 1 and they match the variables and their descriptions in Section 2.1. Can the reviewer give us a more specific idea of where Table 1 does not cross-reference to the earlier text? Secondly, we have added estimated ARIs for four case studies in NZ-- Matata, Thames, Queenstown and Ligar Bay. These are cited in the text (new Table 1) and used as the basis for our analysis. This reinforces the point about a lack of information and awareness, certainly in NZ. |
| 335 | Is this because we can't always capture dynamic risk parameters (e.g., exposure and evacuation with heavy rainfall)? It would be good to highlight the need for dynamic risk models. | OK—but again, we want to focus on the initial demonstration of potential risk to life. The subsequent investigation will not get past first base unless we can demonstrate that there is a problem! |

---

## Referee Report (RR1)

Thank you for addressing the previous questions and comments.

On my second reading of this manuscript, I was thinking about where this methodology may fit within the context or workflow regional scale – debris fan identification and susceptibility mapping. It was only in the conclusion where I found the important and well-expressed point that even after a regional susceptibility assessment has been undertaken "there is a danger that their risk may be considered negligible". It would be helpful for this to be woven throughout the narrative of the text with one or two additional sentences in appropriate parts of the manuscript.

Given the need for precautionary but realistic assumptions, the analysis should state that with an assumption of $P_{s:h} = 1$, the risk values represent a better estimate/upper estimate for the fan apex rather than the fan periphery. Was there any thought given to running two examples, such as the Fan Apex and Fan Periphery from which a range of risk results could be calculated? This approach could help mitigate concerns that the analysis is overly pessimistic, especially since in the manuscript it is stated that $P_{s:h} = 1$ is a "very conservative approach", while also emphasizing the importance of not doing so in risk analysis elsewhere in the risk analysis.

Additionally, it was unclear which risk value is being compared against the individual risk criteria. Are you comparing the individual risk value or the "object risk to people in buildings"? If it is the latter, also known as Total Risk, this cannot be used to compare against risk threshold criteria. Summing probabilities together, as is done with the multiplication of risk by E, does not yield a probability and therefore cannot be compared directly against criteria. Instead, it can be used for relative comparisons, where low sums (less than about 10^-3) are roughly equivalent to societal risk calculations. This concept is supported by binomial approximation.

**Specific Comments**

*Line numbers refer to tracked changes manuscript version*

Line 49: Remove colloquial reference of "landslips"

Line 56: Delete sentence starting "In contrast, potentially catastrophic….". I don't think this sentence is needed and the same issues of non-recognition for debris flows apply to other landslide types

Once again, thank you for considering these suggestions. The revisions you've made have already improved the manuscript, and I appreciate the effort you've put into refining it further. I recommend that the manuscript be accepted subject to minor revisions

---

## Referee Report (RR2)

Review: Identifying unrecognised risks to life from debris flows - September 2024

**Overview:**

Thank you to the authors for submitting a revision of their manuscript which includes several important improvements, including a title and abstract that more closely reflects their work, improved organization, clarification about the sources of landslide data cited in this study, and improved clarity regarding the analysis of theoretical landslide ARIs. The overall writing and organization is well done. The primary conclusion of the paper is that there are likely debris flow fans in New Zealand where there exists unacceptable but as-yet unidentified risk to life from debris flows. This conclusion is based on the fact that the annual probability of debris flows that would result in unacceptable risk falls within a range of debris flows reported in some cases of published debris flow return intervals in the region. While this is a valid observation that could potentially be used to justify further research in the region, in my view it still falls short of a truly meaningful contribution to knowledge. I encourage the authors to continue refining the conclusions of the paper to push an interesting thought exercise into a form that offers impactful conclusions and improve the intellectual merit of the work.

The authors then present a methodological framework that could be useful for identifying areas where risk from debris flows may exist, even if no landslides or debris flows have been documented. While the results of this exploratory analysis are interesting, I am unconvinced that the paper presents a framework that would be applicable by practitioners or that would improve upon informal, ad hoc assessments that a dwelling or community may be exposed to risk of debris flows. This is because:
1. Bayesian analysis of posterior distributions are quick and low-cost, but are not accessible to most risk managers or practitioners without advanced technical expertise and analytical skills. This could be mitigated by providing a map or look-up table for different settlement/records scenarios in their community.
2. The minimum acceptable ARI is estimated with a risk equation that is only very coarsely parameterized for the entirety of New Zealand. See the "Risk Estimate Parameters" section below.

The idea the authors present to use the length of no-observation as to contain posterior ARI distributions is an important one, and some expansion on this idea could support a meaningful contribution to the scientific literature. To improve the applicability of the product and the intellectual contribution of this paper, the authors might consider generating a look-up table that a practitioner could use to estimate likely ARI based on the length of "no landslide" observation, along with guidelines for parametrizing risk based on local observations like the number of houses or relative location to the active debris flow channel.

Alternatively, how could this work be expanded on to provide spatial granularity of risk estimate based on housing density?

**Risk Estimate Parameters:**

I disagree with some of the parameter values the authors have chosen to estimate risk, as described in my first review. I understand that the authors wish to present a simple metric, but I worry that coarse parameter estimates that are difficult to justify and with no sensitivity analysis are likely to result in misleading estimates of risk and ARI, which may be off by an order of magnitude. This is particularly important given the presentation as a methods paper that the authors suggest would be applicable to other regions or for more detailed analysis. I will defer to the editor on the best approach here, but have outlined my primary concerns with some of the parameter estimates here.

- Maximum acceptable risk to life (0.001) - I don't have an issue with this specific value, but it is important that the authors acknowledge that risk tolerance varies greatly according to community values and priorities, and that practitioners could adjust this value for local analysis. For example, some communities may need to balance risk from one natural hazard against another, or may have different risk tolerance for sensitive populations (children or elders). Consider consulting the literature on risk tolerance to add to this discussion. (E.g., Wachinger et al., 2010; Enright 2014, and others). This could be a simple acknowledgement in the paper but is important for future research or practitioners.
- Probability of impact to a dwelling (1.0) - There is almost certainly less than 100% probability that a debris flow would impact a dwelling on a debris flow fan. In fact, the authors undermine their estimate when they describe that 100% probability is credible *if* certain conditions are met. If they have to qualify the probability, then 100% is not appropriate. I recognize that it may be difficult to estimate this value and that the authors wish to choose a conservative value, but they need to explore a range of reasonable values or some other justification, such as an estimate of the average inundation proportion of a debris flow fan during a single event.
- Probability of individual death if an impact occurs (0.1) - I recognize the regional variability in this value, but in my experience the fatality rate is much higher (at least half of impacted debris flows that I have seen that impacted buildings resulted in at least one fatality). To align with your desire for a conservative value, consider using the upper estimate you cite from Glade and others (0.25).
- Dwellings / catchment (1) - This value could easily be off by one or more orders of magnitude. The authors follow an approach from Bell and Glade (2004) to estimate individual risk, which may then be multiplied on the back end, but I would like to see a more explicit analysis of how a larger number of dwellings would change the minimum ARI and discussion in the rest of the paper.

**Minor Comments:**

Overall the language and writing is readable and accurate. I appreciate the clear writing style. Some of the line-by-line comments from the first two reviews have not yet been addressed, such as the colloquial use of the term "landslips" and clarifying "*recorded* history of debris flow" in the manuscript text. Please review the minor comments to ensure that they have all been addressed.

**Recommended Literature:**
- Wachinger, et al., 2010, Risk Perception of Natural Hazards
- Enright, 2014, Is there a tolerable level of risk from natural hazards in New Zealand?